# Genetic determinants facilitating the evolution of resistance to carbapenem antibiotics

**Peijun Ma[1,2,3,4], Lorrie L He[1], Alejandro Pironti[1], Hannah H Laibinis[1], Christoph M Ernst[1,2,3,4], Abigail L Manson[1], Roby P Bhattacharyya[1,5], Ashlee M Earl[1], Jonathan Livny[1], Deborah T Hung[1,2,3,4]\***

[1]The Broad Institute of MIT and Harvard, Cambridge, United States; [2]Department of Molecular Biology, Massachusetts General Hospital, Boston, United States; [3]Department of Genetics, Harvard Medical School, Boston, United States; [4]Center for Computational and Integrative Biology, Massachusetts General Hospital, Boston, United States; [5]Division of Infectious Diseases, Massachusetts General Hospital, Boston, United States

**Abstract** In this era of rising antibiotic resistance, in contrast to our increasing understanding of mechanisms that cause resistance, our understanding of mechanisms that influence the propensity to evolve resistance remains limited. Here, we identified genetic factors that facilitate the evolution of resistance to carbapenems, the antibiotic of 'last resort', in *Klebsiella pneumoniae*, the major carbapenem-resistant species. In clinical isolates, we found that high-level transposon insertional mutagenesis plays an important role in contributing to high-level resistance frequencies in several major and emerging carbapenem-resistant lineages. A broader spectrum of resistance-conferring mutations for select carbapenems such as ertapenem also enables higher resistance frequencies and, importantly, creates stepping-stones to achieve high-level resistance to all carbapenems. These mutational mechanisms can contribute to the evolution of resistance, in conjunction with the loss of systems that restrict horizontal resistance gene uptake, such as the CRISPR-Cas system. Given the need for greater antibiotic stewardship, these findings argue that in addition to considering the current efficacy of an antibiotic for a clinical isolate in antibiotic selection, considerations of future efficacy are also important. The genetic background of a clinical isolate and the exact antibiotic identity can and should also be considered as they are determinants of a strain's propensity to become resistant. Together, these findings thus provide a molecular framework for understanding acquisition of carbapenem resistance in *K. pneumoniae* with important implications for diagnosing and treating this important class of pathogens.

**\*For correspondence:**
dhung@broadinstitute.org

**Competing interests:** The authors declare that no competing interests exist.

## Introduction

Antibiotic resistance is one of the most urgent threats to public health. Resistance has emerged to almost all clinically used antibiotics and in nearly all bacterial pathogen species. Numerous studies have focused on identifying and characterizing resistance mechanisms; meanwhile, our understanding of mechanisms that facilitate the evolution of resistance in clinical isolates is less well understood (*MacLean and San Millan, 2019*). As such, antibiotic efficacy as reflected in minimum inhibitory concentrations (MICs) remains almost the sole criterion to guide clinical antibiotic choice. However, more sophisticated antibiotic stewardship could help to preserve the existing arsenal of antibiotics by better matching individual strains to the antibiotic selected for treatment to minimize the frequency of the evolution of resistance during antibiotic exposure. Such stewardship would need to be informed by an increased understanding of the mechanisms that may affect the evolution of

resistance including microbial intrinsic factors such as the genetic background of an isolate and extrinsic factors such as the antibiotic choice. Practically, understanding these mechanisms would result in the identification of genetic markers of a higher likelihood for resistance evolution, which could usher in a new era of more comprehensive diagnostics to guide more sophisticated antibiotic stewardship.

Bacteria acquire antimicrobial resistance through horizontal gene transfer (HGT) or mutation, processes that can be influenced by intrinsic microbial genetic factors, such as phage defense systems and error prone polymerases, respectively (*Marraffini and Sontheimer, 2008*; *Rosenberg, 2001*). While HGT involves the acquisition of new resistance genes, mutation of existing genes can occur by acquisition of single-nucleotide polymorphisms (SNPs), insertions, deletions, recombination, or transposition events. At the same time, microbe extrinsic factors such as the antibiotic identity can also affect the evolution of resistance, as they vary in their ability to induce mutagenesis (*Cirz et al., 2005*), have different barriers to resistance (*Blázquez et al., 2018*), and vary in their spectrum of possible resistance-conferring mutations.

The carbapenems, which are the latest generation of β-lactams, are often used to treat infections resistant to almost all antibiotics including extended-spectrum β-lactam antibiotics (*Papp-Wallace et al., 2011*; *Zhanel et al., 2007*). Carbapenem resistance thus typically emerges in bacteria that already carry extended-spectrum β-lactamases (ESBLs) and/or other β-lactamases (*Cerqueira et al., 2017*; *Ma et al., 2018*; *Poirel et al., 2007*). Carbapenem resistance is most often mediated by the production of carbapenemases. In the absence of carbapenemases however, resistance can be achieved through the acquisition of a combination of porin mutations to impede drug entry and/or significant increases in β-lactamase expression (*Cerqueira et al., 2017*; *Ma et al., 2018*; *Poirel et al., 2007*; *Martínez-Martínez et al., 1999*). Therefore, the evolution of carbapenem resistance often involves complex mechanisms of HGT and mutation acquisition.

The Gram-negative pathogen *K. pneumoniae* is one of the most prevalent carbapenem-resistant Gram-negative species (*Navon-Venezia et al., 2017*; *Wyres et al., 2020a*). Within this species, carbapenem resistance occurs predominantly in a few clonal groups (CG), such as CG258, CG15, and CG20 (*Cerqueira et al., 2017*; *Wyres et al., 2020a*; *DeLeo et al., 2014*; *Bowers et al., 2015*; *Pitout et al., 2015*; *Munoz-Price et al., 2013*; *Wyres et al., 2020b*). While clonal spread plays a role in the dissemination of carbapenem resistance (*Cerqueira et al., 2017*; *Bowers et al., 2015*), the emergence of new highly resistant lineages (*Rada et al., 2020*; *Marsh et al., 2019*; *Bonnin et al., 2020*; *Mathers et al., 2011*; *Strydom et al., 2020*; *Peirano et al., 2020*) and the independent acquisition of carbapenem resistance by distinct CG258 strains (*DeLeo et al., 2014*; *Marsh et al., 2019*; *Chen et al., 2014*; *Eilertson, 2019*) suggest that ongoing evolution of carbapenem resistance also plays an important role. These observations suggest that the underlying genetic background of CG258 and other emerging lineages may contribute to a higher propensity for resistance acquisition. Recently, many bioinformatic studies have reported that a major phage defense system, the CRISPR-Cas system, is absent in Sequence Type ST258 and ST11 strains, two major lineages of CG258 (*Li et al., 2018*; *Mackow et al., 2019*; *Tang et al., 2020*). As one of the earliest lineages causing outbreaks of carbapenem resistance, ST258 *K. pneumoniae* isolates are responsible for the global spread of *K. pneumoniae* carbapenemases (KPC) (*Cerqueira et al., 2017*; *Bowers et al., 2015*). Therefore, it has been suggested that the lack of CRISPR-Cas systems could be one of the genetic factors contributing to the high rates of carbapenem resistance in this group. However, the more recently emerging lineages, such as ST15 and ST307, do contain such systems and so carbapenem resistance more generally cannot be explained so simply.

Meanwhile, antibiotic identity may also affect the frequency of evolving resistance. Currently, four different carbapenems are available in an intravenous formulation (*Papp-Wallace et al., 2011*): imipenem, meropenem, ertapenem, and doripenem. In addition, faropenem, a related oral antibiotic in the penem class, is available but only outside of the USA (*Gandra et al., 2016*). Although the five drugs share similar structures and mechanisms of action, differences in their pharmacokinetics (ertapenem can be administered once a day while the other carbapenems require administration three to four times per day), stability against β-lactamase hydrolysis, and penicillin-binding protein target preference (*Zhanel et al., 2007*; *Queenan et al., 2010*; *Kohler et al., 1999*; *Sutaria et al., 2018*) may influence the evolution of resistance differently. For example, previous studies have shown that compared to other carbapenems, ertapenem is more susceptible to hydrolysis by some β-lactamases and its cell entry is more impeded by the loss of porins (*Tsai et al., 2017*; *Jones et al., 2005*), raising

the possibility that a broader spectrum of mutations on β-lactamase or porin genes may selectively affect ertapenem but not the other carbapenems.

In this study, to understand how bacterial genetic background and different carbapenems affect the rates of resistance evolution, we compared mutation frequencies (previously defined as the frequency of independent resistant mutants emerging in a given population [*Martinez and Baquero, 2000*]) of carbapenem-susceptible *K. pneumoniae* clinical isolates from 10 lineages and found that isolates from the dominant and emerging carbapenem-resistant lineages had higher mutation frequencies leading to carbapenem resistance than other lineages. We demonstrated that the higher mutation frequencies are caused by high-level transposon insertional mutagenesis, a process leading to resistance gene duplication and reversible porin disruption. We also showed experimentally that one of the major phage defense systems, CRISPR-Cas systems, indeed can play a role in restricting resistance gene acquisition when corresponding spacers sequences are present. Furthermore, we found that a broad spectrum of resistance-conferring mutations for selected carbapenems such as ertapenem contributed to increased resistance rates; importantly, these mutations selected from ertapenem exposure could serve as stepping-stones to high-level resistance to all carbapenems. Taken together, this work identified multiple factors that facilitate the evolution to carbapenem resistance in *K. pneumoniae* clinical isolates and demonstrated that the evolution of antibiotic resistance can be a complex process with important implications for antibiotic selection tailored to the genetic background of clinical isolates.

## Results

### The evolution of carbapenem resistance was affected by genetic background of the isolates

We analyzed genomes of 267 previously sequenced *K. pneumoniae* clinical isolates (*Cerqueira et al., 2017*) and selected carbapenem-susceptible isolates from 10 lineages (*Figure 1A*, *Table 1*, *Supplementary files 1* and *2*). We chose isolates from the predominant carbapenem-resistant lineage that has caused many outbreaks since 2000 (UCI38 [ST258]), the dominant ESBL-producing lineage that is becoming increasingly carbapenem resistant (MGH222 [ST15]), newly emerging carbapenem-resistant lineages (UCICRE126 [ST147], MGH66 [ST29], BIDMC41 [ST37], MGH74 [ST76], MGH158 [ST152], UCI64 [ST17]), and lineages that have not caused carbapenem-resistant clonal outbreaks (UCI34 [ST34] and MGH21 [ST111]). We measured mutation frequencies of these isolates under ertapenem (*Figure 2A*) or rifampicin treatment (*Figure 2B*), using a modified Luria–Delbrück system in which low numbers of bacterial cells were seeded into each well of 384-well plates, thus making the emergence of two independent mutants in the same well extremely unlikely (*Gomez et al., 2017*; *Figure 1B*). This format requires that all resistance occurs through mutation acquisition and not HGT. (We define resistance as at least a twofold increase in the MIC for the mutant relative to the MIC against the original susceptible parent strain, and not relative to the clinically defined MIC breakpoints of the antibiotic. Therefore, resistant mutants selected from our experiments do not necessarily have MICs that are greater than the clinical breakpoints.) We found that except for MGH66 (ST29), all isolates showed similar levels of mutation frequencies to rifampicin (*Figure 2B*), whereas a wide range of mutation frequencies to ertapenem were observed (*Figure 2A*). In particular, some strains had much higher mutation frequencies to ertapenem than to rifampicin. Since resistance to rifampicin is acquired through point mutation resulting from errors during DNA replication (*Goldstein, 2014*; *Pope et al., 2008*), these results suggest that other genetic mechanisms help to determine the mutation frequency to ertapenem.

Among all strains tested, UCI38 (ST258) had the highest mutation frequency to ertapenem. It carries an ESBL gene $bla_{SHV-12}$ on the plasmid pESBL (*Figure 2C*), raising the possibility that ESBL activity could contribute to high-level mutation frequencies. To test this hypothesis, we transformed pSHV (*Figure 2D*), a multi-copy laboratory plasmid containing $bla_{SHV-12}$, amplified from pESBL, into three isolates lacking an ESBL gene and with baseline low-level mutation frequencies to ertapenem, including UCI64 (ST17), UCI34 (ST34), and MGH21 (ST111). However, introduction of $bla_{SHV-12}$ did not change the mutation frequencies of these strains for ertapenem (*Figure 2E*), even though the expression of $bla_{SHV-12}$ was higher in strains transformed with pSHV than in UCI38, which naturally

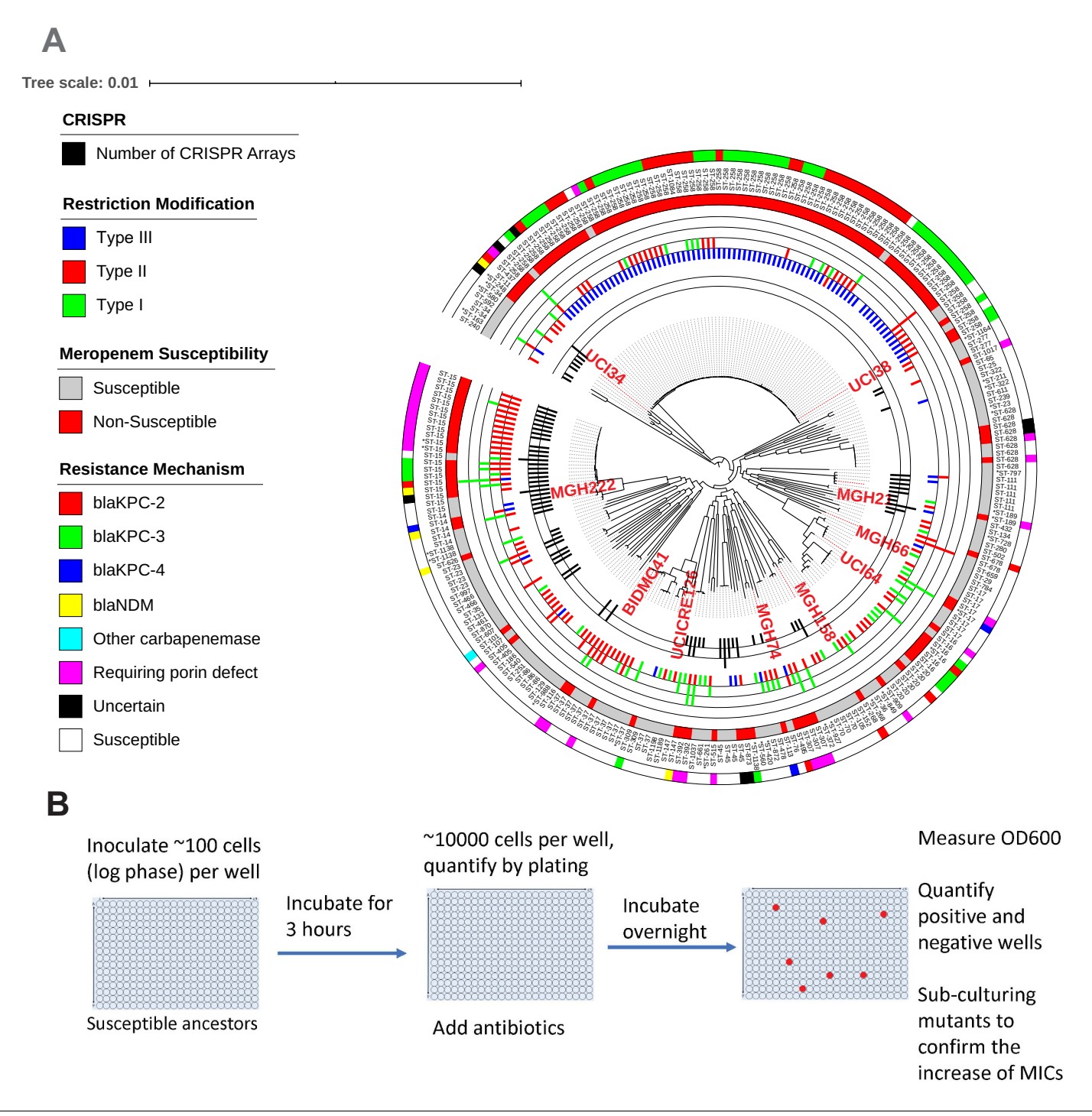

**Figure 1.** Ten phylogenetically diverse carbapenem-susceptible *K. pneumoniae* isolates were selected from a collection of 267 *K. pneumoniae* clinical isolates. (**A**) The selected isolates are highlighted in red. In this phylogenetic tree, from inner to outer circles, the content of the CRISPR-Cas systems, restriction–modification systems, susceptibility to carbapenems, and sequence types are indicated. For carbapenem-resistant isolates, the resistance mechanism is also indicated. (**B**) Scheme of the modified Luria–Delbrück system. Exponential-phase growing cells are diluted and inoculated into 384-well plates, followed by incubation at 37°C for 3 hr. Antibiotics were then added at the concentrations of 1.1× MICs or at specified concentrations, and cultures were incubated at 37°C overnight. OD600 was measured the next day, and positive and negative wells were quantified. Mutants from each plate were sub-cultured in MHB medium supplemented with the same antibiotics at the same concentrations used for the selection, and saved in 25% glycerol stocks for future analysis. Mutants that did not grow up in the sub-culturing were excluded from the calculation of mutation frequencies.

**Table 1.** Genetic features of 10 selected carbapenem-susceptible clinical isolates.

| Strains | ST | MIC (µg/ml) of | | Narrow-spectrum β-lactamase | ESBL | Carbapenemase | *ompK36* | Accession no. | CRISPR system | R-M system |
|---|---|---|---|---|---|---|---|---|---|---|
| | | ETP | MEM | | | | | | | |
| UCI38 | ST258 | 0.25 | 0.06 | None | $bla_{SHV-12}$ | None | Intact | GCA_000566805.1 | No | Type II; Type III |
| MGH222 | ST15 | 0.03 | 0.03 | None | $bla_{SHV-28}$ | None | Intact | GCA_014902955.1 | Yes | None |
| UCICRE126 | ST147 | 0.03 | 0.25 | $bla_{SHV-11}$ | None | None | Intact | GCA_014902315.1 | Yes | Type I |
| MGH66 | ST29 | 0.06 | 0.03 | $bla_{SHV-187}$ | None | None | Intact | GCA_000694555.1 | No | Type II |
| BIDMC41 | ST37 | 2 | 0.06 | $bla_{TEM-1}$ | $bla_{SHV-12}$ | None | Intact | GCA_000492195.1 | No | Type II |
| MGH74 | ST76 | 0.03 | 0.06 | $bla_{SHV-1}$ | None | None | Intact | GCA_000694715.1 | No | Type I; Type II |
| MGH158 | ST152 | 0.03 | 0.06 | $bla_{SHV-1}$ | None | None | Intact | GCA_002152555.1 | No | None |
| UCI64 | ST17 | 0.03 | 0.03 | $bla_{SHV-11}$ | None | None | Intact | GCA_000688175.1 | No | Type I |
| UCI34 | ST34 | 0.06 | 0.03 | $bla_{SHV-26}$ | None | None | Intact | GCA_000566845.1 | Yes | None |
| MGH21 | ST111 | 0.03 | 0.03 | $bla_{SHV-11}$ | None | None | Intact | GCA_000492915.1 | Yes | None |

carries $bla_{SHV-12}$ (*Figure 2—figure supplement 1*). This ruled out the simple presence of the ESBL gene alone as the reason for the differing mutation frequencies.

Next, we sought to test the hypothesis that the whole plasmid, pESBL (*Figure 2C*), might confer high-level mutation frequencies to ertapenem. However, when we attempted to transform pESBL into the same three strains with low-level mutation frequencies to ertapenem, none of them could take up pESBL. In contrast, an ST258 strain BWH41 (the only ST258 isolate lacking an ESBL gene in our collection) and a laboratory strain of *E. coli*, 10β, could take up pESBL (*Figure 2F*). Meanwhile, all strains successfully took up pSHV with similar efficiencies, suggesting that pESBL was uniquely restricted in particular strains under regular laboratory conditions.

## A type I-E CRISPR-Cas system prevented the acquisition of antibiotic resistance genes via HGT, while other genetic factors contribute to high mutation frequencies

To understand why pESBL is restricted in these three isolates but not BWH41 (ST258), we analyzed the genomic sequences of the collection of 267 *K. pneumoniae* isolates for the presence of two major phage defense systems, the CRISPR-Cas systems and restriction–modification (R-M) systems (*Supplementary file 3*), which function to exclude foreign DNA. We found that of the three strains which could not take up pESBL, MGH21 (ST111), and UCI34 (ST34) have type I CRISPR-Cas systems, while UCI64 (ST17) has no CRISPR-Cas system but carries type I R-M systems. In contrast, among 80 strains of the ST258 lineage, we found no CRISPR-Cas systems and most strains carry type III R-M system (*Figure 1A* and *Supplementary file 3*). When we broadened our analysis to include the genomic sequences of 2453 *K. pneumoniae* strains available in the NCBI database, including 550 ST258 strains, we found that no ST258 strains contain a CRISPR-cas system (*Supplementary file 4*), confirming that the lack of CRISPR-Cas system is a genetic feature of the ST258 lineage. This finding is consistent with other bioinformatic studies which have tried to link the absence of CRISPR systems in ST258 strains to carbapenem resistance (*Li et al., 2018*; *Mackow et al., 2019*; *Tang et al., 2020*). However, there is no clear association between the absence of CRISPR and the more recently emerging carbapenem-resistant lineages (*Figure 1A* and *Supplementary files 3* and *4*).

To understand the ability of MGH21 (ST111) to restrict pESBL uptake, a strain that encodes a type I-E CRISPR-Cas system but no R-M systems, we first confirmed by RNA sequencing (RNA-seq) that indeed the CRISPR-Cas system was expressed in MGH21 (*Figure 3—figure supplement 1*). We then compared the sequence of pESBL with MGH21's CRISPR-Cas system and found that MGH21 has a spacer (spacer 11) (*Figure 3A* and *Supplementary file 5*), targeting a gene encoding a DNA-methyltransferase (DNMT) (*Bujnicki and Radlinska, 1999*) in pESBL (*Figure 2C*); by searching a curated plasmid database (*Brooks et al., 2019*), we found that this spacer additionally aligns with

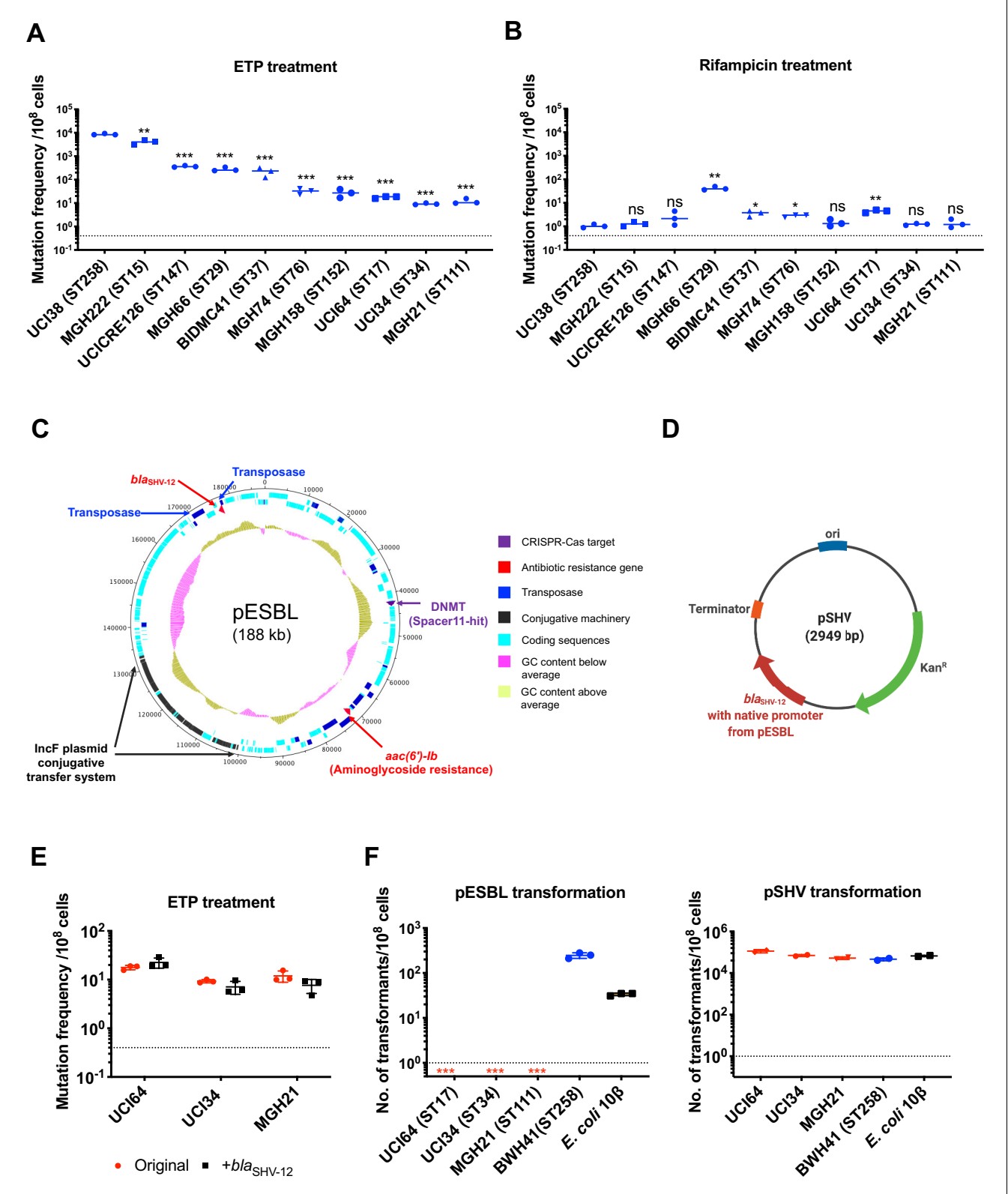

**Figure 2.** The evolution of carbapenem resistance is affected by genetic background of the isolates. (**A**) Mutation frequencies of 10 clinical isolates under treatment with ertapenem. Five isolates, UCI38 (ST258), MGH222 (ST15), UCICRE126 (ST147), MGH66 (ST29), and BIDMC41 (ST37), have relatively greater mutation frequencies to ertapenem (>100 mutants per $10^8$ cells) than the other five isolates. Comparing to UCI38 (ST258) that has the highest mutation frequencies to ertapenem, all isolates have significantly different mutation frequencies. Two-tailed Student's t-test was used for statistical

*Figure 2 continued on next page*

*Figure 2 continued*

analysis between UCI38 (ST258) and other isolates. (B) Mutation frequencies of 10 clinical isolates under treatment with rifampicin. Isolates with relatively high-level mutation frequencies to ertapenem do not necessarily have high-level mutation frequencies to rifampicin. Two-tailed Student's t-test was used for statistical analysis between UCI38 (ST258) and other isolates. (C) Diagram of pESBL, an ESBL-encoding plasmid isolated from UCI38 (ST258). (D) Diagram of pSHV, a multi-copy laboratory plasmid containing the native promoter and coding region of the ESBL gene $bla_{SHV-12}$ amplified from pESBL. (E) The ESBL gene, $bla_{SHV-12}$, was amplified from pESBL and expressed in three isolates lacking an ESBL gene and with relatively low-level mutation frequencies to ertapenem. However, mutation frequencies to ertapenem were not changed compared to the original strains lacking an ESBL gene (red). Two-tailed Student's t-test was used for statistical analysis to compare the original strain with the corresponding strain overexpressing $bla_{SHV-12}$, with p>0.05 for all three pairs. (F) Transformation efficiencies of pESBL (left) or pSHV (right) in three isolates lacking ESBL genes (red) and with relatively low-level mutation frequencies to ertapenem. As controls, these two plasmids were also transformed into another ST258 strain BWH41 (blue), which does not carry ESBL genes, and a strain of *E. coli* 10β (black). pESBL could not be transformed into these three isolates but it could be transformed into BWH41 (ST258) and *E. coli*. In contrast, the laboratory construct pSHV was successfully transformed into all strains tested. For all experiments in (A, B, E, F) two to three independent biological replicates were performed. Data from independent experiments were plotted individually with error bars plotted as the standard deviation. The limit of detection is indicated with a dashed line, and the asterisk (*) under the dashed line indicates frequencies under the limit of detection. *p<0.05; **p<0.005; ***p<0.0005; ns, not significant.

The online version of this article includes the following figure supplement(s) for figure 2:

**Figure supplement 1.** Relative expression of $bla_{SHV-12}$ in three strains overexpressing $bla_{SHV-12}$ through pSHV.

sequences found in an additional 94 other plasmids carrying antibiotic resistance genes, including 62 multi-drug resistance plasmids (plasmids carrying resistance genes to more than one class of antibiotics) and 21 plasmids carrying carbapenemase genes (*Supplementary file 6* and *7*). In addition, spacer24 (*Figure 3A* and *Supplementary file 5*) aligned to a conserved hypothetical gene that was also found in UCI38 as well as 66 additional plasmids carrying antibiotic resistance genes, including 44 multi-drug resistant plasmids and 12 plasmids carrying carbapenemase genes (*Supplementary file 6* and *8*). Collectively, these results pointed to the potential role of the CRISPR-Cas system in excluding the uptake of resistance carrying plasmids such as pESBL. Indeed, after depleting the CRISPR-Cas operon (MGH21Δ*cas*; the CRISPR-Cas system along with two adjacent hypothetical genes was deleted), pESBL could now be successfully transformed, whereas episomal complementation of the CRISPR-Cas system back into MGH21Δ*cas* again restricted pESBL transformation (*Figure 3B*).

Unsurprisingly, the absence of CRISPR-Cas system increased rates at which resistance by HGT could be acquired but did not change mutation frequencies of MGH21 (*Figure 3D*). In contrast, introduction of pESBL into MGH21Δ*cas* increased the frequency with which resistance to ertapenem emerged in our modified Luria–Delbruck system where HGT cannot occur; the frequency for MGH21Δ*cas* (pESBL) was ~30 times higher than for the parent MGH21, MGH21Δ*cas*, or MGH21 carrying pSHV (*Figure 3D*). As introduction of the ESBL gene alone in pSHV does not change resistance frequencies, this elevation suggests that factors on pESBL other than the ESBL gene contributed to the high mutation frequencies. Furthermore, while MGH21Δ*cas*(pESBL) had elevated ertapenem mutation frequencies relative to MGH21, its frequency was still 10–20 times lower than that of UCI38 itself, from which pESBL was isolated (*Figure 3D*), suggesting that differences between the genetic backgrounds of MGH21 and UCI38, irrespective of pESBL, play additional roles in high frequency mutation acquisition.

## Transposon insertional mutagenesis caused frequent and reversible inactivation of porin genes leading to ertapenem resistance

To gain insight into other genetic factors that may cause the different levels of mutation frequencies to ertapenem between UCI38 and MGH21, we analyzed whole genome sequencing (WGS) data of laboratory-derived resistant mutants to identify the specific genetic events leading to ertapenem resistance. We compared six ertapenem resistant mutants derived from UCI38 (ST258), five mutants derived from MGH21 (ST111), and five mutants derived from MGH21Δ*cas*(pESBL) (*Figure 4A*). We found that the two strains carrying pESBL favored transposition events as a mechanism to attain resistance while the strain lacking pESBL, MGH21, developed resistance only through SNP acquisition. All six resistant mutants derived from UCI38 were due to duplication of the transposon on pESBL in which the $bla_{SHV-12}$ is embedded (*Figure 2C*) and/or disruption of *ompK36*, one of the major porin genes of *K. pneumoniae* that facilitates carbapenem cell entry, by insertion sequences

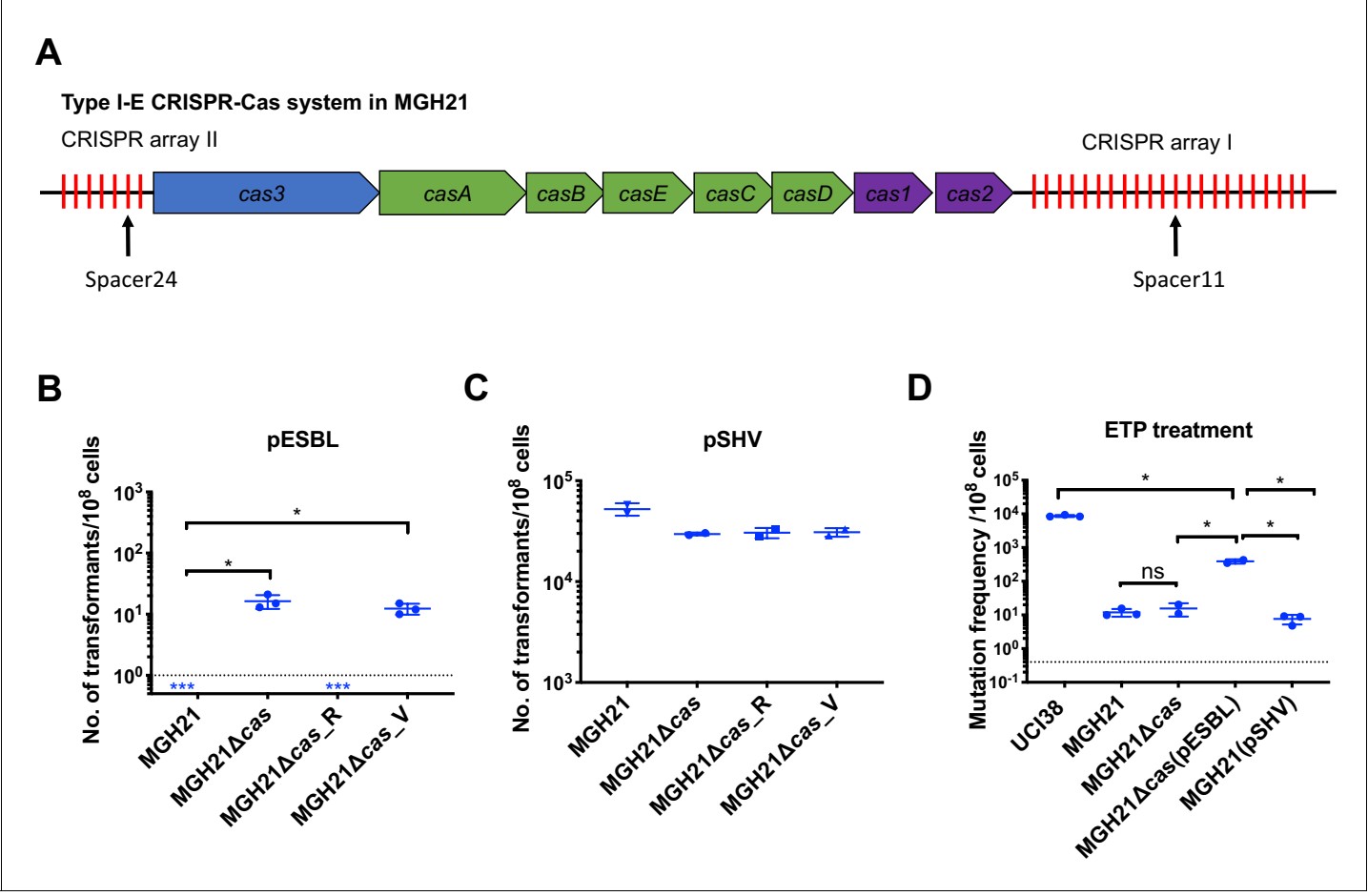

**Figure 3.** Type I-E CRISPR-Cas system in MGH21 (ST111) prevents the acquisition of pESBL but the presence of pESBL alone does not account for high mutation frequencies. (A) Type I-E CRISPR-Cas system in MGH21 (ST111). The two CRISPR arrays and the position of two spacers (Spacer11 and Spacer24) that align to plasmids encoding resistance genes are indicated. Spacer11 aligns to the DNMT gene located on pESBL _(*Figure 2C*). (B, C) Transformation efficiencies of pESBL (B) or the lab construct pSHV (C) in MGH21, MGH21Δ*cas*, MGH21Δ*cas*_R with CRISPR-Cas complementation, and MGH21Δ*cas*_V with control vector complementation. pESBL could only be transformed into MGH21 strains in which the CRISPR-Cas system was deleted (MGH21Δ*cas* and MGH21Δ*cas*_V), whereas pSHV could be transformed into all strains at similar efficiencies. (D) Mutation frequencies of UCI38 (ST258), MGH21, MGH21Δ*cas*, MGH21Δ*cas*(pESBL), and MGH21(pSHV) with ertapenem treatment. The deletion of the CRISPR-Cas system (MGH21Δ*cas*) and the introduction of pSHV (MGH21(pSHV)) did not affect the mutation frequencies. In contrast, the introduction of pESBL (MGH21Δ*cas* (pESBL)) increased mutation frequencies, indicating that some factors on pESBL other than the ESBL gene affect the mutation frequencies. However, mutation frequencies of MGH21Δ*cas*(pESBL) were still significantly lower than these of UCI38, indicating that more factors in the genetic background of UCI38 contribute to the high-level mutation frequencies. All experiments were performed in triplicate and data were plotted individually. Error bars were plotted as standard deviation. The limit of detection of each assay is indicated with a dashed line, and the asterisk (*) under the dashed line indicates that the transformation efficiencies are below the limit of detection. Two-tailed Student's t-test was used for all statistical analysis; an asterisk marking a pair-wise comparison denotes a p<0.05.

The online version of this article includes the following figure supplement(s) for figure 3:

**Figure supplement 1.** RNA-seq data shows that *cas* genes and most spacers are expressed in MGH21.

---

(ISs, small transposons that only carry the transposase genes). (Although the other porin OmpK35 also facilitates cell entry for carbapenems, we found no resistant mutants carrying mutations in *ompK35*, probably due to the low expression levels of *ompK35* in the growth condition used (*Nicolas-Chanoine et al., 2018*; *Figure 4—figure supplement 1*) or pre-existing mutations already disrupting *ompK35* (*Bowers et al., 2015*) in some strains.) Similarly, for MGH21Δ*cas*(pESBL), four mutants stemmed from the same transposon duplication of *bla*_SHV-12 on pESBL, while the fifth mutant resulted from the acquisition of a SNP in *ompK36*. In contrast, all resistant mutants derived from MGH21 resulted from the acquisition of SNPs or short deletions/insertions, mostly in porin

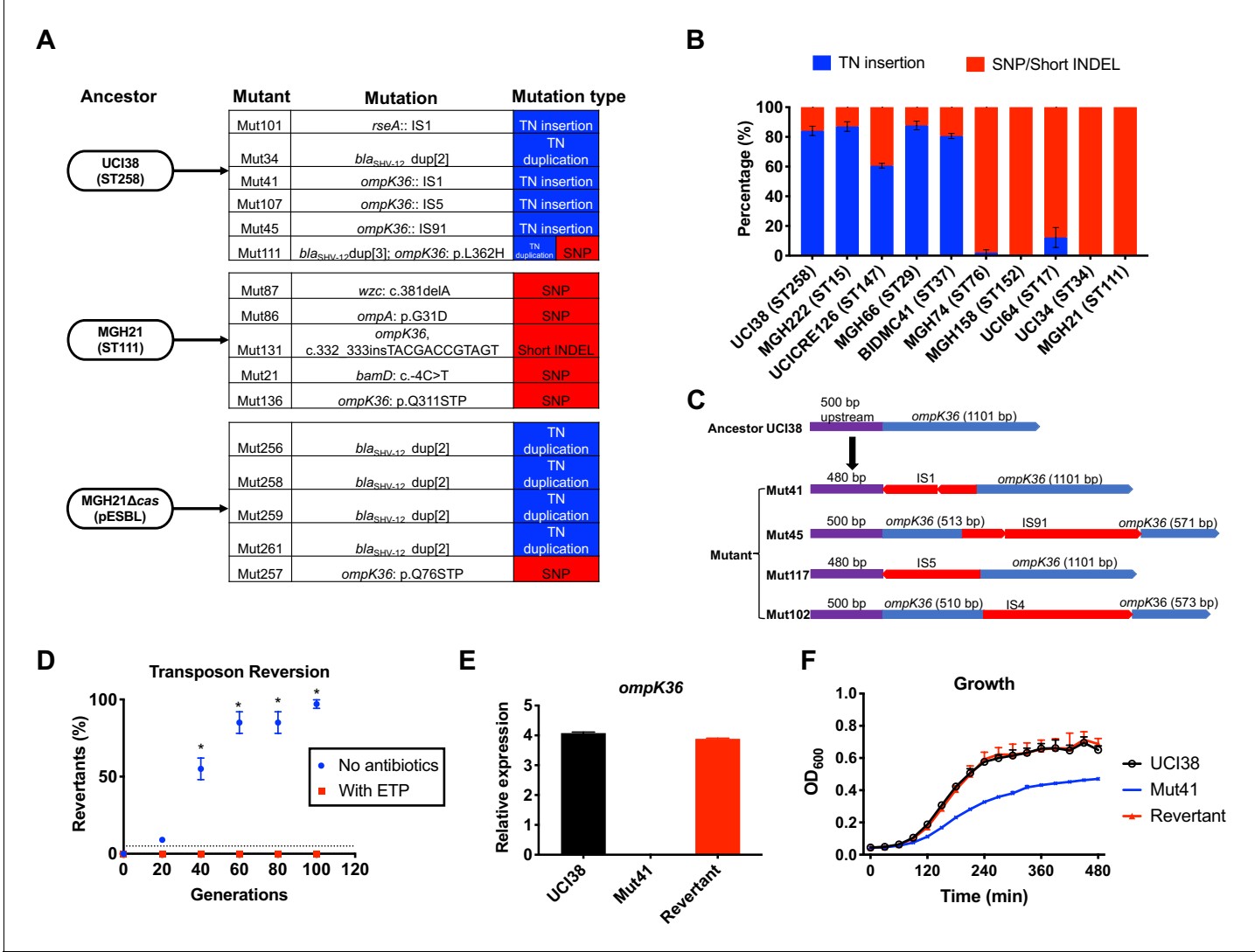

**Figure 4.** Transposon insertional mutagenesis causes frequent and reversible inactivation of porin genes in isolates with high-level mutation frequencies to ertapenem. (**A**) Mutation types (transposon insertion/duplication [blue] vs. SNP [red]) identified via WGS in ertapenem resistant mutants of UCI38 (ST258), MGH21 (ST111), and MGH21Δcas(pESBL). The majority of mutants that carry pESBL (UCI38 and MGH21Δcas(pESBL)) have transposon-mediated mutations, while only SNPs or short insertion/deletions were observed in mutants of MGH21. (**B**) Relative quantification of the propensity of 10 selected isolates to undergo transposon insertion (blue) versus SNP acquisition (red) in *ompK36* during ertapenem treatment. For each of these strains, 50–100 mutants were isolated, and the types of mutation in *ompK36* locus, if any, was determined via Sanger sequencing. Transposon insertions occurred at ~10 times higher frequencies than the acquisition of SNPs or short insertion/deletion in strains with higher level of mutation frequencies to ertapenem. (**C**) Illustration of *ompK36* inactivation by four transposons in UCI38. Four representative mutants derived from UCI38 were selected and the insertion sites were determined by Sanger sequencing. (**D**) Transposon disruption of *ompK36* was reversible. A representative ertapenem resistant mutant with an IS1 insertion in *ompK36*, Mut41, was cultured in the presence or absence of ertapenem. Every 20 generations, colony PCR targeting *ompK36* locus was performed on the culture to quantify the percentage of the population that had lost the transposon insertion at this locus. Two-tailed Student's t-test was used for statistical analysis at each time point to compare the cultures with and without antibiotics. (**E, F**) The relative expression of *ompK36* (**E**) and growth curves (**F**) of UCI38 (black), Mut41 (red), and one representative revertant of Mut41 (blue). All experiments were performed in triplicate. Error bars are plotted as the standard deviation.

The online version of this article includes the following figure supplement(s) for figure 4:

**Figure supplement 1.** Relative expression of *ompK36* and *ompK35*.

**Figure supplement 2.** Scheme of the assay for quantification of transposon insertions and SNPs in *ompK36*.

**Figure supplement 3.** Copy number of ISs in each strain.

**Figure supplement 4.** Reversion of TN-insertion mutants derived from UCI22, MGH66 and BIDMC41.

genes and outer membrane protein genes. pESBL thus increased mutation frequencies relative to pSHV because $bla_{SHV-12}$ on pESBL lies within a transposon that can be easily duplicated to elevate ESBL expression and thus MIC (*Figure 3D*). In contrast, while carrying pSHV intrinsically conferred a higher baseline MIC because of its higher $bla_{SHV-12}$ expression level (*Figure 2—figure supplement 1*), it could not duplicate $bla_{SHV-12}$ to further evolve increased MIC, thus explaining its unchanged mutation frequencies relative to the parent MGH21 (*Figure 2E*).

Comparing the two strains that carry pESBL, we noted that UCI38 was able to disrupt *ompK36* through transposon insertion, while MGH21Δ*cas*(pESBL) only did so through SNP acquisition. We hypothesized that the higher likelihood of a disrupting transposition event rather than the acquisition of a disrupting SNP might explain the higher mutation frequencies of UCI38 and other strains with relatively high-level mutation frequencies to ertapenem (*Figure 2A*). Indeed, when we used the modified Luria-Delbrück system to isolate and characterize 50–100 ertapenem resistant mutants from each of these 10 isolates (*Figure 4—figure supplement 2*; *Supplementary file 9*), we found that transposon insertions in *ompK36* accounted for 60–90% of resistant mutants derived from strains with high-level mutation frequencies to ertapenem, while only 0–10% of mutants resulted from transposon insertion in *ompK36* in strains with relatively lower mutation frequencies to ertapenem (*Figure 4B*). Of note, no one specific IS element accounted for the high transposition rates, as ISs from four different families (IS4, IS5, IS91, IS1) were involved in the inactivation of *ompK36* (*Figure 4D* and *Supplementary file 10*). There was also no correlation between the number of ISs and the activity-level of transposon insertional mutagenesis (*Figure 4—figure supplement 3*). Nevertheless, these results demonstrate that a higher propensity for transposon insertional mutagenesis in some genetic backgrounds was an important contributor to the more facile evolution of ertapenem resistance in some strains, with such events occurring at nearly 10 times higher frequency than SNP acquisition.

In contrast to SNP acquisition for which a reversion is extremely rare, transposon insertions can be reversible (*Mahillon and Chandler, 1998*). Since porin disruption is known to come at a fitness cost in the absence of antibiotic selective pressure (*Knopp and Andersson, 2015*; *Phan and Ferenci, 2017*), the mechanism of transposon disruption of *ompK36* to achieve antibiotic resistance in UCI38 afforded a potentially facile path, i.e., reversion, to recover from this fitness cost when selective pressure is removed. Indeed, this reversion was observed when we cultured Mut41 (*Figure 4C*), a mutant of UCI38 carrying an IS1 insertion in the promoter region of *ompK36*, without antibiotics (*Figure 4D*). Ninety-nine percent of the population reverted to the wild-type *ompK36* gene by ~100 generations, thereby restoring both the expression of *ompK36* and the fitness of the strain relative to the parent mutant Mut41 (*Figure 4E,F*). We observed the same phenomenon in mutants derived from three other strains (*Figure 4—figure supplement 4*), demonstrating the high versatility of this resistance mechanism. A high propensity for transposon insertional mutagenesis resulting in porin inactivation provides a fitness advantage in the presence of antibiotic, while preserving a path to restoration of fitness in the absence of antibiotics.

## Spectrum of genetic mutations conferring resistance to ertapenem is broader than to meropenem

Next, we explored how different carbapenems affect the rates at which resistance evolves. We measured mutation frequencies in response to treatment with four carbapenems and faropenem in three representative carbapenem-susceptible *K. pneumoniae* clinical isolates: UCI38 (an ST258 strain carrying one chromosomal ESBL $bla_{SHV-12}$ and a second episomal $bla_{SHV-12}$ copy), MGH21 (an ST111 strain with a single copy of the non-ESBL $bla_{SHV-11}$ on the chromosome), and MGH32 (an ST111 strain with no β-lactamase genes because the single native, chromosomal $bla_{SHV-1}$ is inactivated) (*Figure 5* and *Supplementary file 1*). The lowest mutation frequencies resulted from meropenem treatment while relatively higher frequencies resulted from ertapenem and faropenem. In the case of MGH32, which carries no β-lactamase gene, we did not isolate resistant mutants to any of the carbapenems including ertapenem, but isolated resistant mutants to faropenem (*Figure 5*), indicating that β-lactamase genes may be necessary for the evolution of resistance to carbapenems, but not to faropenem. To confirm that our observation was not limited to these three strains, we measured mutation frequencies of an additional three isolates under separate treatment of these five antibiotics, and similar patterns were observed, suggesting that the influence of carbapenem identity is independent of the genetic background of strains (*Figure 5—figure supplement 1*).

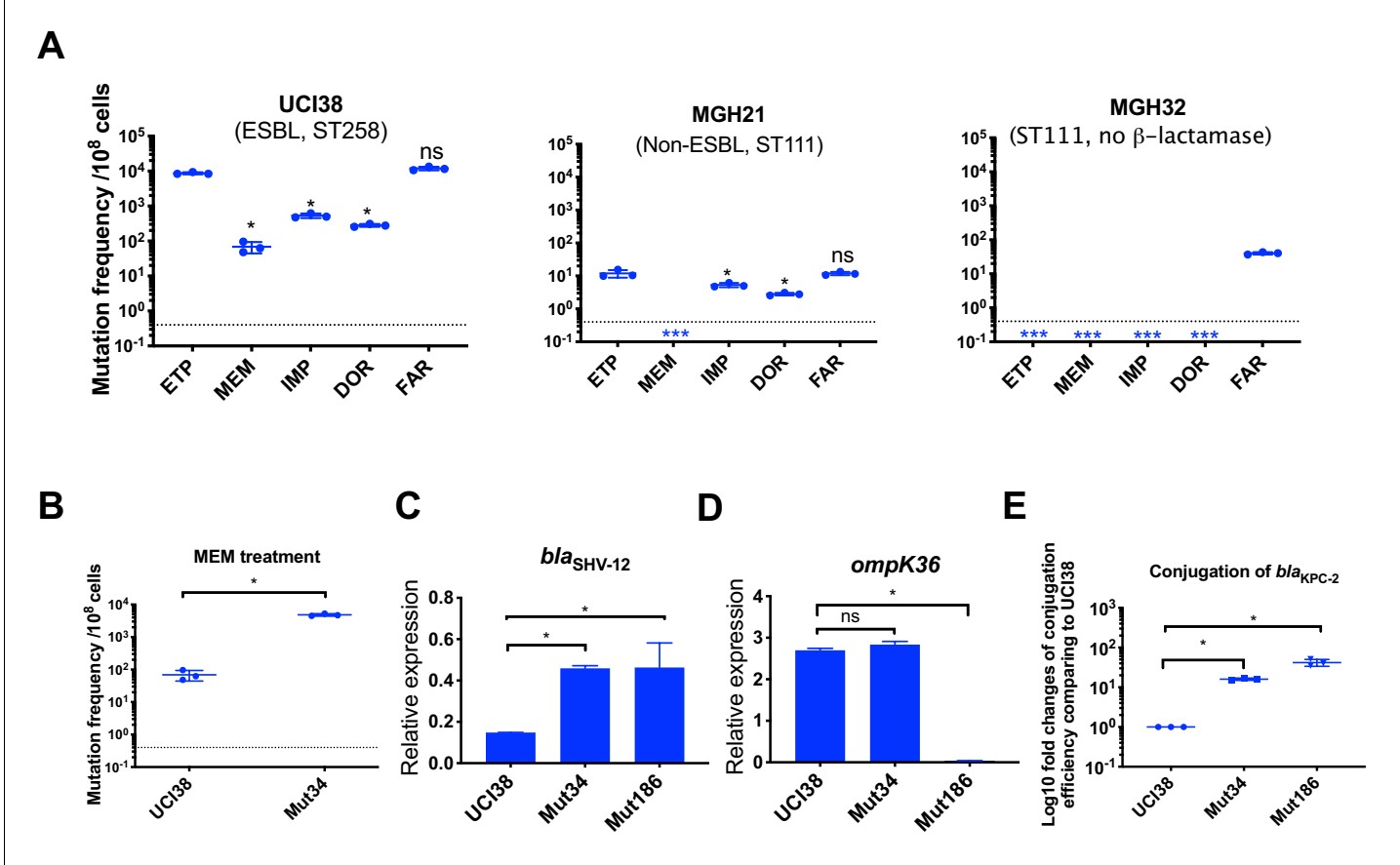

**Figure 5.** Ertapenem and faropenem treatment are not only associated with higher mutation frequencies, but they also promote the evolution of meropenem resistance. (A) Mutation frequencies of three representative isolates, UCI38 (ST258), MGH21 (ST111), and MGH32 (ST111, no β-lactamase), under separate treatment with ertapenem (ETP), meropenem (MEM), imipenem (IMP), doripenem (DOR), or faropenem (FAR). Higher mutation frequencies are associated with ertapenem and faropenem treatment, while lower mutation frequencies are observed with meropenem treatment. In MGH32, an isolate without β-lactamase genes, only faropenem resistant mutants were isolated. Two-tailed Student's t-test was used for statistical analysis to compare between ertapenem treatment and other carbapenems or faropenem. (B) Mutation frequencies of UCI38 and Mut34, an ertapenem-restricted-resistant mutant derived from UCI38, under treatment with meropenem. Despite having the same MIC of meropenem as UCI38, Mut34 had higher mutation frequencies than UCI38. (C, D) Relative expression levels of $bla_{SHV-12}$ (C) or $ompK36$ (D) in UCI38, Mut34, and Mut186 (an ertapenem and meropenem-resistant mutant derived from Mut34) show the progressive acquisition of mutations to achieve meropenem resistance. Mut34 has increased $bla_{SHV-12}$ relative to its parent UCI38; Mut186 has disrupted $ompK36$, relative to its parent Mut34. (E) Conjugation efficiencies of UCI38, Mut34, and Mut186 with *K. pneumoniae* clinical isolate BIDMC45 carrying $bla_{KPC-2}$. In the presence of meropenem, Mut186 had the highest conjugation efficiency with UCI38 having the lowest. All experiments were performed in triplicate. Two-tailed Student's t-test was used for statistical analysis to compare UCI38 with other strains. Error bars are plotted as standard deviation. The limit of detection is indicated with a dashed line, and the asterisk (*) under the dashed line indicates frequencies under the limit of detection.

The online version of this article includes the following figure supplement(s) for figure 5:

**Figure supplement 1.** Mutation frequencies of three other isolates under separate treatment with ertapenem (ETP), meropenem (MEM), imipenem (IMP), doripenem (DOR), or faropenem (FAR).

**Figure supplement 2.** Higher mutation frequencies associated with ertapenem were not due to stability of these drugs or to the induction of mutagenesis.

**Figure supplement 3.** Conjugation efficiencies of UCI38, Mut34, and Mut186 (derived from Mut101 with meropenem treatment) with a *K. pneumoniae* clinical isolate BIDMC45 carrying $bla_{KPC-2}$.

**Figure supplement 4.** Prior exposure to faropenem promotes the evolution of meropenem resistance.

Because ertapenem and meropenem were equally stable under these assay conditions (*Figure 5— figure supplement 2*), and bacteria were treated with concentrations of antibiotic normalized to their MICs for each drug, the different mutation frequencies were not explained by differences in antibiotic exposure. We also ruled out the possibility that ertapenem could induce more mutagenesis than meropenem, a phenomenon that has been described for some β-lactams (*Miller et al.,*

*2004*), by measuring the mutation frequencies to rifampin after pre-treatment with sub-MIC concentrations of ertapenem, meropenem, or ciprofloxacin (a fluoroquinolone antibiotic known to induce mutagenesis [*Cirz et al., 2005*]) as a positive control. While both carbapenems increased rifampin mutation frequencies compared with untreated controls, each did so quivalently, and less than ciprofloxacin (*Figure 5—figure supplement 2B*).

We then turned to the possibility that ertapenem's higher mutation frequency could be due to a greater spectrum of resistance-conferring mutations than for meropenem. We isolated and characterized 90 mutants, derived from UCI38 or MGH21, that were selected from our modified Luria-Delbrück system with confirmed shifts in the corresponding MICs of ertapenem and meropenem (*Table 2* and *Supplementary file 11*). Sixty-three mutants had increases in the MICs, relative to their corresponding ancestor strains, of both ertapenem (2- to 256-fold increases) and meropenem, albeit with relatively lower levels of meropenem resistance (2- to 16-fold increases). We did not isolate any mutants that are highly resistant (MIC > 4 µg/ml) to meropenem. Meanwhile, 27 mutants only had corresponding increases in the MICs of ertapenem, and not meropenem (*Supplementary file 11*). No mutants had an increased MIC of meropenem but not ertapenem.

We analyzed WGS data from 10 representative mutants, five that had MIC shifts to both ertapenem and meropenem, and five that had MIC shifts only to ertapenem (*Table 2*), and validated all identified resistance-conferring mutations by complementation (*Supplementary file 12*). Six of the mutants contained either transposon insertions or SNPs in *ompK36* or duplication of *bla*SHV-12. Interestingly, four mutants carried novel mutations, including mutations in *wzc* (capsule synthesis), *ompA* (porin), *rseA* (anti-sigma E factor), and the promoter region of *bamD* (outer membrane protein assembly factor), with the first three resulting in selective ertapenem resistance. These results show that indeed ertapenem had a wider allowable spectrum of resistance-conferring mutations than meropenem, which yielded a higher mutation frequency.

## Pre-selection with ertapenem increased the likelihood of evolving resistance to meropenem by both spontaneous mutation and HGT

While many ertapenem-resistant mutants do not display resistance to meropenem, we found that acquisition of such mutations, while not impacting the immediate efficacy of meropenem as reflected in the MIC, impacted its future efficacy by increasing the frequency at which resistance to

**Table 2.** Characterization of representative mutants resistant to both ertapenem and meropenem or to ertapenem alone.

| Mutant ID | Ancestor | Mutations causing decreased susceptibility | Gene function | Fold changes of MICs compared to ancestor strains | |
|---|---|---|---|---|---|
| | | | | ETP | MEM |
| Mutants resistant to ertapenem alone | | | | | |
| Mut87 | MGH21 | *wzc* (L367_02683): c.381delA | Exopolysaccharide biosynthesis | 2 | 1 |
| Mut86 | MGH21 | *ompA* (L367_001084): p.G31D | Porin | 2 | 1 |
| Mut131 | MGH21 | *ompK36* (L367_002817), c.332_333insTACGACCGTAGT | Porin | 256 | 1 |
| Mut101 | UCI38 | *rseA* (P841_001338):: IS1 | Anti-sigma E factor, involved in maintaining cell envelope integrity | 8 | 1 |
| Mut34 | UCI38 | *bla*SHV-12 (P841_005417) dup[2] | ESBL | 8 | 1 |
| Mutants resistant to ertapenem and meropenem | | | | | |
| Mut21 | MGH21 | *bam*D (L367_003146): c.-4C > T | Outer membrane protein assembly factor | 8 | 2 |
| Mut136 | MGH21 | *omp36K (L367_002817)*: p.Q311STP | Porin | 64 | 8 |
| Mut41 | UCI38 | *ompK36* (P841_001022):: IS1 | Porin | 64 | 8 |
| Mut107 | UCI38 | *ompK36* (P841_001022):: IS5 | Porin | 64 | 16 |
| Mut45 | UCI38 | *ompK36* (P841_001022):: IS91 | Porin | 64 | 16 |

meropenem emerges. The mutation frequencies of an ertapenem-restricted resistant strain (Mut34, which carries a duplication of $bla_{SHV}$ on pESBL [*Table 2*]) were more than 100 times greater than the frequency of its corresponding parental strain UCI38 under identical meropenem treatment (*Figure 5B*). WGS of the meropenem-resistant mutants revealed that the majority of the mutants derived from Mut34 had acquired new mutations in the porin gene *ompK36* (i.e., Mut186, *Figure 5C,D*), to accompany the previously acquired $bla_{SHV-12}$ duplication. These results demonstrate that the previously acquired mutation in Mut34 that confers ertapenem resistance alone could serve as a stepping-stone to the subsequent acquisition of a porin-disrupting mutation to yield meropenem resistance.

Of note, de novo mutation acquisition, even in this stepping-stone fashion, resulted in only low to moderate levels of meropenem resistance (4- to 32-fold increase in MIC from the ancestor strains). With the hypothesis that HGT of carbapenemases or additional ESBL genes may be required to evolve truly high-level meropenem resistance, we examined the impact of the ertapenem-limited resistance mutations on the ability to horizontally acquire resistance genes. Indeed, in the presence of meropenem, higher rates of uptake of a clinical plasmid carrying the carbapenemase gene $bla_{KPC-2}$ were observed for both Mut34 and Mut186 than the ertapenem-sensitive parental strain UCI38 (*Figure 5E*); rather than a direct mechanistic impact, this finding is likely due to longer survival times of these mutants in the presence of meropenem compared to the parental strain affording them a greater opportunity to pick up the plasmid, as the conjugation frequencies are the same in the absence of meropenem (*Figure 5—figure supplement 3*). A faropenem-limited-resistant mutant, Mut101, like Mut34 for ertapenem, also showed elevated mutation frequencies and conjugation efficiencies in the presence of meropenem compared to its parental strain (*Supplementary file 1* and *Figure 5—figure supplement 4*). Together these results suggest that ertapenem and faropenem not only elicit more frequent resistance themselves, but they also select for mutations that can increase the rates at which bacteria acquire high-level meropenem resistance.

## Discussion

In this study, we identified genetic factors that facilitate the evolution of carbapenem resistance in *K. pneumoniae* clinical isolates (*Figure 6*), one of the most alarming antibiotic-resistant pathogens that have emerged due to our limited arsenal against such organisms. We find that high-level transposon insertional mutagenesis and the mutational spectrum for each carbapenem play important roles in increased mutation frequencies. These mutational mechanisms can work in conjunction with loss of systems that restrict horizontal resistance gene uptake, i.e., the CRISPR-Cas systems, to facilitate the evolution of resistance.

We found that isolates of major and emerging carbapenem-resistant lineages indeed have high-level mutation frequencies to carbapenem antibiotics compared to lineages that have not been linked to carbapenem resistance; this is due to high-level transposon insertional mutagenesis in lineages associated with carbapenem resistance. This highlights the notion that the emergence of predominant resistant lineages did not occur through random events and provide genetic markers that signal isolates with high risk of developing resistance. Importantly, this mechanism of acquiring resistance could serve an evolutionary advantage as the disruption of porins by transposons can revert (*Figure 4D*), thereby enabling strains to rapidly adapt to fluctuating environments and optimizing their survival in the presence and absence of antibiotic exposure. The fact that many of the more recently emerging lineages, such as ST15 and ST307, have evolved resistance by a combination of ESBLs and porin truncations may potentially point to the relevance of such mutagenic mechanisms. More generally, transposon-mediated gene duplication has been reported to contribute to heteroresistance in many different bacterial species and antibiotic classes (*Nicoloff et al., 2019*; *Andersson et al., 2019*). This study thus provides further evidence that mutational events mediated by transposons play a critical role in the evolution of antibiotic resistance in parallel with HGT.

Bioinformatic studies have previously suggested a potential relationship between the absence of CRISPR-Cas systems and carbapenem resistance in the ST258 lineage (*Li et al., 2018*; *Mackow et al., 2019*; *Tang et al., 2020*). However, as the more recent resistant lineages to emerge still retain CRISPR-Cas systems, the absence of such systems cannot fully explain the emergence of resistance. Here we demonstrated that they indeed can play a role in restricting the uptake of resistance plasmids, if accompanied by appropriate spacers (*Figure 3*). Importantly, bioinformatic

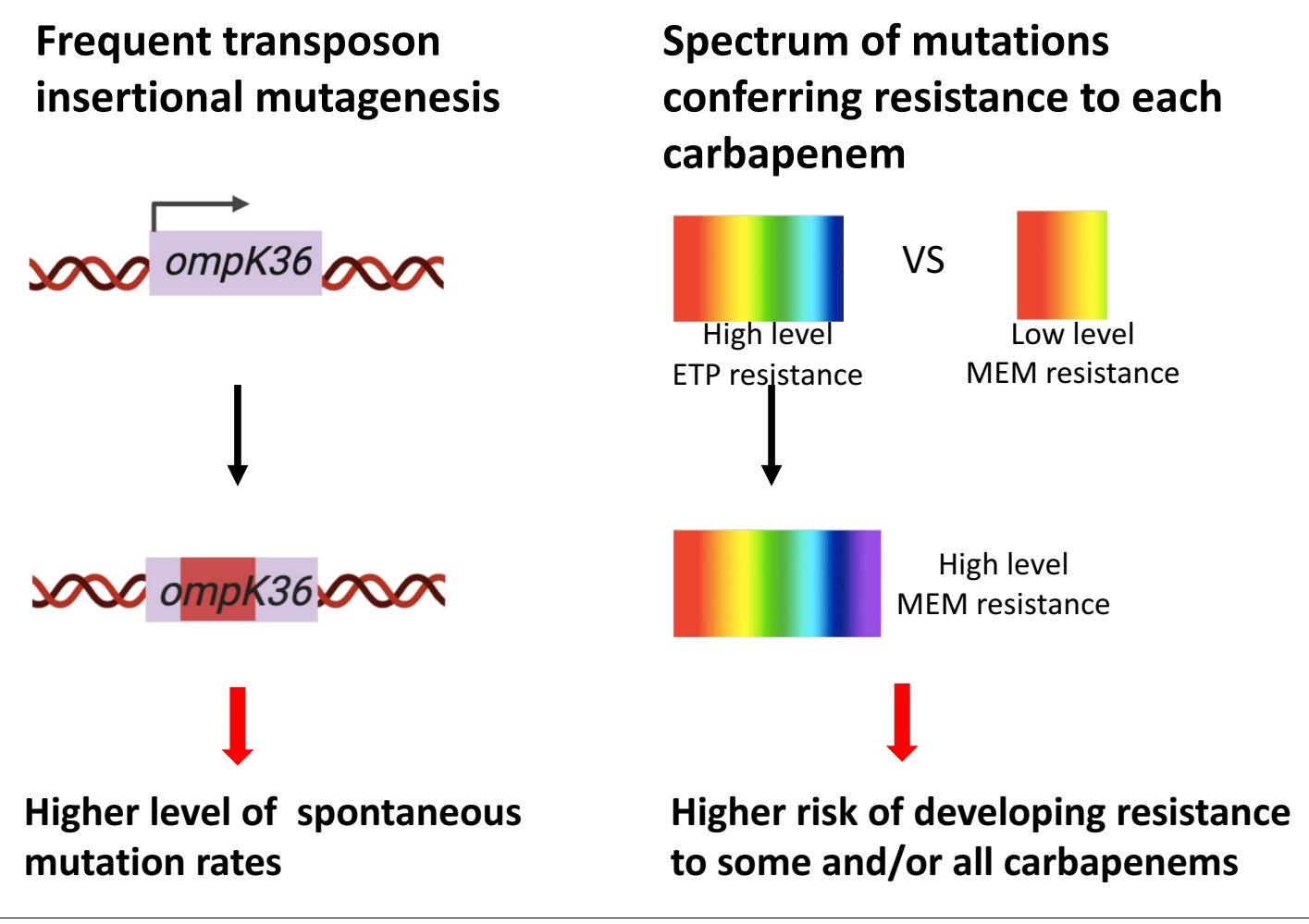

**Figure 6.** Two genetic determinants of the evolution of carbapenem resistance were identified from this study. On the one hand, high-level transposon insertional mutagenesis facilitates the inactivation of porin genes. On the other hand, a broader spectrum of genetic mutation conferring resistance to ertapenem leads to higher rates of developing resistance with ertapenem treatment; these ertapenem-restricted resistance mutations can serve as stepping-stones to facilitate the development of high-level resistance to all carbapenems.

analysis of spacer sequences, and not simply the presence or absence of a CRISPR-Cas system alone, is needed to understand the functional role of such systems in resistance gene exclusion in the recently resistant lineages.

The mutational spectrum that confers resistance to each carbapenem also affects evolution frequencies. Currently in practice, several factors affect the choice of a specific carbapenem or faropenem in treating a patient, including its availability, spectrum of activity, dosing schedule, route of administration, and cost. Ertapenem is sometimes favored for the convenience of its once-daily dosing, whereas the other three carbapenems all require three to four doses per day. However, ertapenem and faropenem lack activity against *Pseudomonas aeruginosa*, thus limiting their use in some infections (*Zhanel et al., 2007*; *Rodríguez-Baño et al., 2018*). Besides these factors, mutation frequencies associated with these antibiotics have not been taken into consideration in antibiotic prescription. In this study, we show that a higher resistance frequency is associated with ertapenem and faropenem due to the broader spectrum of resistance-conferring mutations than is allowed for other carbapenems such as meropenem. Importantly, these mutations can serve as stepping-stones to facilitate the evolution of high-level resistance to all carbapenems. As ertapenem or faropenem are often favored for the convenience of its once-daily dosing or oral bioavailability, respectively, these results highlight the non-equivalence of antibiotics even within the same class of antibiotics with

respect to the propensity to evolve resistance. It might suggest that the use of carbapenems with a higher barrier to resistance should be favored to prevent the evolution of carbapenem resistance.

Currently, the choice and administration of an antibiotic is based almost solely on the MIC as an indicator of susceptibility. However, this work shows that treating strains with similar MICs with the same antibiotic could have different outcomes with regards to the emergence of resistance. Isolates with diverse genetic backgrounds can have very different mutational frequencies, despite having the same MIC (*Table 1*). Clearly, some genetic mutations pre-selected from ertapenem or faropenem treatment are not sufficient to change MICs of meropenem, but they can significantly increase the likelihood of evolving resistance to meropenem.

This work calls attention to the fact that all strains of the same species are not, and should not be thought of as identical, with regards to their potential contribution to the problem of antibiotic resistance and perhaps infection, more generally. It argues for the potential importance of characterizing strains beyond MIC measurements alone, as part of a new generation of more sophisticated diagnostics, including identifying lineages that have higher propensity to develop resistance (i.e., ST258) and stepping-stone mutations that herald the potential impending emergence of resistance (i.e., mutations in *rseA* and *ompK36*). Understanding and identifying the presence of these mechanisms could have far-reaching implications for antibiotic choice in favor or those with higher barriers to resistance (i.e., meropenem). Future steps that are needed include, importantly, extending these findings into the clinic, which will require tracking ESBL and CRE strains in the clinic to associate them with patient metadata and outcome and to monitor strain evolution after antibiotic exposure. At the same time, new diagnostic technologies would be required to rapidly provide these higher levels of genetic detail, as next-generation sequencing cannot yet meet either the cost or speed requirements for a universally deployed diagnostic modality. Clear algorithms would then need to be devised to dictate antibiotic selection based on the genetic background of strains to minimize the emergence of resistance. A final implication is that consideration of the relative barriers to resistance should be prioritized in the development of subsequent generations of same-in-class antibiotics to ensure that no agent becomes widely available that would erode the efficacy of the entire class.

In this current era of rising antibiotic resistance, as significant investment is needed in the discovery of new antibiotics, parallel efforts are needed to guide more judicious use of our current available antibiotics to minimize the emergence of resistance. This work suggests that strategies should not only consider current efficacy, but also consider both the genetic backgrounds of strains and antibiotic choice as they impact the potential for erosion of future efficacy. More generally, this work demonstrates that investigating evolutionary drivers of antibiotic resistance can reveal the root causes of resistance evolution, thereby providing a framework to improve current clinical diagnosis and antibiotic selection.

## Materials and methods

**Key resources table**

| Reagent type (species) or resource | Designation | Source or reference | Identifiers | Additional information |
|---|---|---|---|---|
| Strain, strain background (*K. pneumoniae*) | UCI38 | *Cerqueira et al., 2017* | GCA_000566805.1 | A *K. pneumoniae* clinical isolate |
| Strain, strain background (*K. pneumoniae*) | MGH222 | *Cerqueira et al., 2017* | GCA_014902955.1 | A *K. pneumoniae* clinical isolate |
| Strain, strain background (*K. pneumoniae*) | UCICRE126 | *Cerqueira et al., 2017* | GCA_014902315.1 | A *K. pneumoniae* clinical isolate |
| Strain, strain background (*K. pneumoniae*) | MGH66 | *Cerqueira et al., 2017* | GCA_000694555.1 | A *K. pneumoniae* clinical isolate |

*Continued on next page*

*Continued*

| Reagent type (species) or resource | Designation | Source or reference | Identifiers | Additional information |
|---|---|---|---|---|
| Strain, strain background (*K. pneumoniae*) | MGH74 | *Cerqueira et al., 2017* | GCA_000694715.1 | A *K. pneumoniae* clinical isolate |
| Strain, strain background (*K. pneumoniae*) | MGH158 | *Cerqueira et al., 2017* | GCA_002152555.1 | A *K. pneumoniae* clinical isolate |
| Strain, strain background (*K. pneumoniae*) | MGH21 | *Cerqueira et al., 2017* | GCA_000492915.1 | A *K. pneumoniae* clinical isolate |
| Strain, strain background (*K. pneumoniae*) | MGH32 | *Cerqueira et al., 2017* | GCA_000493075.1 | A *K. pneumoniae* clinical isolate in which the native $bla_{SHV-1}$ was inactivated by a point mutation (Leu88STP) |
| Strain, strain background (*K. pneumoniae*) | UCI43 | *Cerqueira et al., 2017* | GCA_000566745.1 | A *K. pneumoniae* clinical isolate |
| Strain, strain background (*K. pneumoniae*) | UCI22 | *Cerqueira et al., 2017* | GCA_000566925.1 | A *K. pneumoniae* clinical isolate |
| Strain, strain background (*K. pneumoniae*) | UCI44 | *Cerqueira et al., 2017* | GCA_000566725.1 | A *K. pneumoniae* clinical isolate |
| Strain, strain background (*K. pneumoniae*) | UCI34 | *Cerqueira et al., 2017* | GCA_000566845.1 | A *K. pneumoniae* clinical isolate |
| Strain, strain background (*K. pneumoniae*) | MGH30 | *Cerqueira et al., 2017* | GCA_000492935.1 | A *K. pneumoniae* clinical isolate |
| Strain, strain background (*K. pneumoniae*) | BIDMC40 | *Cerqueira et al., 2017* | GCA_000492215.1 | A *K. pneumoniae* clinical isolate |
| Strain, strain background (*K. pneumoniae*) | UCI64 | *Cerqueira et al., 2017* | GCA_000688175.1 | A *K. pneumoniae* clinical isolate |
| Strain, strain background (*K. pneumoniae*) | BIDMC41 | *Cerqueira et al., 2017* | GCA_000492195.1 | A *K. pneumoniae* clinical isolate |
| Strain, strain background (*K. pneumoniae*) | BWH41 | *Cerqueira et al., 2017* | GCA_000567545.1 | A *K. pneumoniae* clinical isolate |
| Strain, strain background (*K. pneumoniae*) | MGH21SHV | This study, available upon request | MGH21SHV | MGH21(pSHV), MGH21 expressing $bla_{SHV-12}$ |
| Strain, strain background (*K. pneumoniae*) | MGH21Δ*cas* | This study, available upon request | MGH21Δ*cas* | MGH21 in which the operon encoding the CRISPR-Cas system was deleted |
| Strain, strain background (*K. pneumoniae*) | MGH21Δ*cas*(pESBL) | This study, available upon request | MGH21Δ*cas*(pESBL) | MGH21Δ*cas* carrying pESBL isolated from UCI38 |
| Strain, strain background (*K. pneumoniae*) | MGH21Δ*cas*(pCas) | This study, available upon request | MGH21Δ*cas*(pCas) | MGH21Δ*cas* expressing CRISPR-Cas system through two lab constructs |
| Strain, strain background (*K. pneumoniae*) | MGH21Δ*cas*(pVector) | This study, available upon request | MGH21Δ*cas*(pVector) | MGH21Δ*cas* carrying two empty vectors |

*Continued on next page*

*Continued*

| Reagent type (species) or resource | Designation | Source or reference | Identifiers | Additional information |
|---|---|---|---|---|
| Strain, strain background (*E. coli*) | *E. coli* 10ß | NEB (C3020) | Cat # C3020 | *E. coli* strain from NEB |
| Strain, strain background (*K. pneumoniae*) | Mut34 | This study, available upon request | Mut34 | Mutant of UCI38 from ertapenem treatment, with the $bla_{SHV-12}$ duplication |
| Strain, strain background (*K. pneumoniae*) | Mut101 | This study, available upon request | Mut101 | Mutant of UCI38 from faropenem treatment, *ompK36* is down-regulated due to the inactivation of *rseA* |
| Strain, strain background (*K. pneumoniae*) | Mut186 | This study, available upon request | Mut186 | Mutant of Mut34 from meropenem treatment, with both $bla_{SHV-12}$ duplication and *ompK36* inactivation |
| Strain, strain background (*K. pneumoniae*) | Mut195 | This study, available upon request | Mut195 | Mutant of Mut101 from meropenem treatment, with both $bla_{SHV-12}$ duplication and *ompK36* down-regulation |
| Strain, strain background (*K. pneumoniae*) | UCI38R | This study, available upon request | UCI38R | Rifampin resistant version of UCI38 |
| Strain, strain background (*K. pneumoniae*) | Mut34R | This study, available upon request | Mut34R | Rifampin resistant version of Mut34 |
| Strain, strain background (*K. pneumoniae*) | Mut101R | This study, available upon request | Mut101R | Rifampin resistant version of Mut101 |
| Strain, strain background (*K. pneumoniae*) | Mut186R | This study, available upon request | Mut186R | Rifampin resistant version of Mut186 |
| Strain, strain background (*K. pneumoniae*) | Mut195R | This study, available upon request | Mut195R | Rifampin resistant version of Mut195 |
| Strain, strain background (*K. pneumoniae*) | BIDMC45 | *Cerqueira et al., 2017* | GCA_000567025 | A *K. pneumoniae* clinical isolate carrying $bla_{KPC-2}$ |
| Recombinant DNA reagent | pSHV (plasmid) | This study, available upon request | pSHV | $bla_{SHV-12}$, including upstream 500 bp, was amplified from UCI38, and blunt ligated into pSmart_LCKN (Kanamycin^R) |
| Recombinant DNA reagent | pKOV (plasmid) | Addgene | RRID:Addgene_25769 | The plasmid used for gene knockout though homologous recombination, Chloramphenicol^R |
| Recombinant DNA reagent | pKOV-*cas*KO (plasmid) | This study, available upon request | pKOV-*cas*KO | A 2 kb DNA fusion containing 1 kb upstream and 1 kb downstream of *cas* operon in MGH21 was ligated into pKOV using BamHI and NotI sites, Chloramphenicol^R |
| Recombinant DNA reagent | pBAD33Gm (plasmid) | This study, available upon request | pBAD33Gm | pBAD33 with Gentamycin resistance, a gentamycin resistance gene was ligated into HindIII cloning site on pBAD33 (Gentamycin^R, Chloramphenicol^R) |
| Recombinant DNA reagent | pBAD33Gm_CasCRISPR1 (plasmid) | This study, available upon request | pBAD33Gm_CasCRISPR1 | *CasABECD*, *cas1*, *cas2* operon, and CRISPR array I was amplified from MGH21, a SD sequence was incorporated upstream of ATG of casA, then this piece of DNA was ligated into KpnI and XbaI sites on pBAD33Gm (Gentamycin^R, Chloramphenicol^R) |

*Continued on next page*

*Continued*

| Reagent type (species) or resource | Designation | Source or reference | Identifiers | Additional information |
|---|---|---|---|---|
| Recombinant DNA reagent | pCas3CRISPR2 (plasmid) | This study, available upon request | pCas3CRISPR2 | *Cas3* gene, including the upstream 500 bp region and CRISPR array II region, was amplified from MGH21 and ligated into pSmart LC KN, Kanamycin[R] |
| Recombinant DNA reagent | pBAD33 (plasmid) | *Guzman et al., 1995* | pBAD33 | Cloning vector, Chloramphenicol[R] |
| Recombinant DNA reagent | pSmart LC KN (vector) | Lucigen | Cat. # 40821 | Cloning vector, Kanamycin[R] |

## Modified Luria–Delbrück experiment

The robotic, modified Luria–Delbrück system was adapted from a system that was previously described by *Gomez et al., 2017*. Exponential growth phase bacterial cultures were diluted to roughly 100 cells per 50 µl (2000 cells/ml) in Mueller–Hinton Broth (MHB) medium. Then the diluted culture was transferred to three to six 384-well microplates (Falcon, cat. # 353962) using Bravo liquid handling platform (Agilent), and each well of these 384-well plates contained 50 µl of the culture. The plates were sealed using BioExcell Film for Tissue Culture (Worldwide life science, cat. # 41061023) and placed in humidified containers at 37°C. After incubating for 3 hr, 10 µl cultures were taken from three randomly selected wells and diluted for plating on Luria–Bertani (LB) agar plates to quantify cell numbers. Then antibiotics at specified concentrations were added to the wells using Bravo liquid handling platform at specified concentrations. After adding antibiotics, cultures were incubated in humidified containers at 37°C overnight. The second-day morning, OD$_{600}$ was read using SpectroMax plate reader (Molecular Device) and mutation frequency was calculated using the following equation: $1 - \sqrt[c]{\left(\frac{w_n}{w_t}\right)}$, where $c$ is the number of cells per well at the time of adding antibiotics, $w_n$ is the number of negative wells, and $w_t$ is the total number of wells. Mutants from each plate were sub-cultured in MHB supplemented with the same antibiotics at the same concentrations used for the selection and saved in 25% glycerol stocks for future analysis. Mutants that did not grow up in the sub-culturing were excluded from the calculation of mutation frequencies. Each experiment was repeated three times.

To measure mutation frequencies with rifampin after carbapenem treatment, exponential growth phase cultures of UCI38 (OD$_{600}$ ~ 0.2) were diluted 100 times with MHB medium, then the diluted cultures were split into four identical cultures. Cell numbers were quantified by plating diluted cultures on LB agar plates, and these cell numbers were used to calculated mutation frequencies. Ertapenem, meropenem, and ciprofloxacin were then added to three of these four cultures at 0.1× MICs. The fourth culture was untreated. Immediately after adding antibiotics, cultures from each condition were aliquoted into three 384-deep-well plates (VWR, cat. # 82051-326) with 50 µl per well using Bravo liquid handling platform. Then these twelve 384-deep-well plates were incubated at 37°C with shaking for 2 hr. After incubating for two hours, 10 µl cultures from each well of these 12 deep-well plates were correspondingly transferred to wells of twelve 384-clear-bottom microplates (Falcon, cat. # 353962), using Bravo liquid handling platform. Then 50 µl MHB supplemented with rifampin at the concentration of 60 µg/ml was added to each of these wells. The final rifampin concentration in each well was 50 µg/ml. These plates were incubated in humidified containers at 37°C overnight. The second-day morning, OD$_{600}$ was read using SpectroMax plate reader (Molecular Device), and mutation frequency was calculated using this equation: $1 - \sqrt[c_0]{\left(\frac{w_n}{w_t}\right)}$, where $c_0$ is the number of cells per well before pre-treatment, $w_n$ is the number of negative wells, and $w_t$ is the total number of wells. This experiment was repeated three times.

## Bacterial strains and plasmid construction

Bacterial strains used in this study are listed in *Supplementary file 1*. All strains were cultured in LB medium or MHB, shaking at 37°C or 30°C as specified.

To construct the plasmid pSHV, $bla_{SHV-12}$, including the 500 base pairs (bp) upstream region, were PCR amplified from UCI38, respectively, using primers listed in *Supplementary file 13*. Then the PCR products were ligated into vector pSmart LC Kn (Lucigen, cat. # 40821) and electroporated into *E. coli* competent cells 10β (NEB, cat. # C3020K). Plasmids were then extracted from positive clones and electroporated into *K. pneumoniae* cells that have been made electroporation competent according to the protocol described previously (*Zheng et al., 2007*). In brief, *K. pneumoniae* cells were streaked on LB agar plates and grown overnight at 37°C. Then cells were collected directly from LB plates and re-suspended in ice-cold sterilized $H_2O$, followed by washing with ice-cold steril-ized $H_2O$ three times. Finally, cells were re-suspended at the concentration of roughly $10^9$ cells/ml for electroporation. Strains expressing $bla_{SHV-12}$ were cultivated in medium supplemented with kana-mycin at the concentration of 25 µg/ml.

To generate MGH21Δ*cas*, about 1000 bp upstream and downstream of the *cas* operon was amplified from MGH21 using Q5 DNA polymerase (NEB, cat. # M0492). Overlap extension PCR was used to fuse these two pieces of DNA to generate a ~ 2000 bp fragment, which was then ligated to pKOV vector (*Link et al., 1997*) using BamHI and NotI sites, resulting in the construct pKOV-*cas*KO. The construct was transformed to *E. coli* competent cells 10β (NEB, cat. # C3020K) via electropora-tion, and the positive transformants were cultured in LB medium supplemented with chloramphenicol (34 µg/ml) at 30°C. Plasmids were then extracted and electroporated into MGH21 electro competent cells and incubated at 30°C on LB agar plates supplemented with chloramphenicol (34 µg/ml) overnight. The integration of the plasmid in either the upstream or the downstream region of the *cas* operon was selected by chloramphenicol resistance and screened by PCR. Following the selection, the integrants were grown in non-selective LB medium for several generations and then plated on LB agar medium with 10% sucrose to induce double recombination. Among the survivors of the sucrose–LB medium, the double recombinants were selected by PCR screening. The deletion of the *cas* operon was confirmed by sequencing and RT-qPCR.

To restore the CRISPR-Cas system to MGH21Δ*cas*, *cas3*, including upstream 500 bp and the CRISPR array II, was amplified from MGH21 and ligated into pSmart LC Kn (Lucigen, cat. # 40821), generating pCas3CRISPR2. Meanwhile, the coding region of *casABECD*, *cas1*, *cas2*, and CRISPR array I was amplified from MGH21 and ligated into pBAD33Gm (*Guzman et al., 1995*) using KpnI and XbaI cloning sites, resulting in pBAD33Gm_CasCRISPR1. A SD sequence was also added 8 bp upstream of ATG codon of *casA*. These two constructs were separately transformed into *E. coli* 10β (NEB, cat. # C3020K) via electroporation. Plasmids were extracted, mixed at 1:1 ratio, and trans-formed into MGH21Δ*cas*, generating the strain MGH21Δ*cas*(pCas). The transformants containing these two constructs were confirmed using PCR and Sanger sequencing. Similarly, the vector control strain MGH21Δ*cas*(pVector) was generated through co-transforming two empty vectors, pSmart LC KN and pBAD33Gm, into MGH21Δ*cas* strain. When mutation frequencies of MGH21Δ*cas*(pCas) and MGH21Δ*cas*(pVector) with ertapenem were measured in MHB medium supplemented with 1% arabi-nose (to induce the expression of *casABECD*), kanamycin (25 µg/ml), and gentamicin (10 µg/ml).

## Plasmids extraction and sequencing from UCI38

Plasmids from UCI38 were extracted using QIAfilter Plasmid Midi Kit (Qiagen, Cat.# 12243). Extracted plasmids were then transformed into other clinical isolates and MGH21Δ*cas* through elec-troporation. Transformants were selected on LB agar plates supplemented with cefotaxime at the concentration of 10 µg/ml. The extracted plasmid DNA was sequenced, assembled and annotated as described before (*Cerqueira et al., 2017*).

## Analysis of 267 *K. pneumoniae* genomes

We used a total of 267 *K. pneumoniae* assemblies generated at the Broad Institute for this analysis, including 80 ST258 strains. *K. pneumoniae* isolates were sequenced, assembled, and annotated as described before (*Cerqueira et al., 2017*). To improve resistance gene predictions, the original gene calls from each assembly were searched against the following databases using BLAST (*Altschul et al., 1990*): (1) Resfinder (*Zankari et al., 2012*) (downloaded January 23, 2018); (2) the National Database of Antibiotic Resistant Organisms (https://www.ncbi.nlm.nih.gov/patho-gens/antimicrobial-resistance/; downloaded January 22, 2018); and (3) an in-house database of car-bapenemases and ESBLs (*Cerqueira et al., 2017*). For each gene, the database hit with the highest

bit score having an e-value $< 10^{-10}$ and gene length coverage $\geq 80\%$ was retained. The numbers of annotated carbapenemases and β-lactamases, including extended-spectrum and broad-spectrum β-lactamases, were quantified and tabulated for each strain.

## Annotation of restriction–modification systems

We downloaded a total of seven reference gene sets for type I (n = 3), type II (n = 2), and type III (n = 2) restriction–modification systems from REBASE (http://rebase.neb.com/rebase/rebase.seqs.html) on May 22, 2019. We used blastn to search for these reference genes in all 267 *K. pneumoniae* assemblies, using an e-value cutoff of $10^{-10}$ and requiring 80% coverage of the reference gene. We retained the top blast hit for each reference gene set and strain. We considered a restriction–modification system of a certain type to be present in a given strain if at least one gene from each of the two (for types II and III) or three (for type I) reference sets were present in the strain.

## Annotation of CRISPR arrays and *cas* genes

CRISPR Detect (*Biswas et al., 2016*) version 2.2 was used to detect CRISPR arrays in the 267 *K. pneumoniae* assemblies and 2453 *K. pneumoniae* strains available in the NCBI database using default parameters. *Cas* genes were identified using the Broad Institute's microbial annotation pipeline. For the CRISPR arrays identified in MGH21, spacer sequences were aligned to a curated database of plasmid sequences (*Brooks et al., 2019*) containing sequences of 6642 plasmids, using blastn and requiring with >80% identity and coverage. Then the sequences of plasmids containing the spacer-hit genes were extracted. ResFinder (*Zankari et al., 2012*) was used to identify antibiotic resistance genes in these plasmids, if any, requiring >95% identity and 80% coverage.

## Determination of MICs

MICs were determined by the broth microdilution method as described (*Wiegand et al., 2008*). The MICs were measured in duplicates in MHB medium, with a final inoculum size of $5 \times 10^5$ cells/ml.

## Quantification of transposon insertions and SNPs in *ompK36*

Following the robotic, modified Luria-Delbrück experiment with ertapenem treatment, 50–100 resistant mutants from each strain were isolated and streaked on LB agar plates supplemented with ertapenem at the concentration of 1.1x MIC against the ancestor strain. Colony PCR was performed using primers listed in *Supplementary file 13* to amplify *ompK36* locus including upstream 500 bp region of each mutant. The PCR products were then purified and Sanger sequenced. Sequences were aligned to the genomic sequences of the ancestor strains and single-nucleotide variants and transposon insertions could thus be quantified.

## WGS and variant calling

Genomic DNA was isolated using DNeasy Blood and Tissue Kits (Qiagen, cat. # 69504) and quantified using Qubit dsDNA HS Assay Kit (Invitrogen, cat. # Q32851). WGS libraries were made using Nextera XT DNA library preparation kit (Illumina, cat. # FC-131–1096). Then the samples were sequenced using the MiSeq or NextSeq system with 300 cycles, pair-ended. For each strain sequencing, depth was set at approximately 100× coverage. BWA mem version 0.7.12 (*Li and Durbin, 2009*) and Pilon v1.23, using default settings (*Walker et al., 2014*), were used to align reads against a reference genome assembly and to identify variants, respectively. SNP positions having mapping quality less than 10 (MQ < 10) were not considered. The *Klebsiella pneumoniae* MGH21, and UCI38 genome assemblies, were used as references for variant identification for mutants derived from each respective strain.

## RNA extraction and RT-qPCR

Cells were cultivated in MHB or LB medium at 37°C until early-exponential growth phase. RNA was purified using Direct-zol RNA Kits (Zymo research, cat. # R2070) and quantified with Nanodrop spectrophotometer (ThermoFisher). RT-qPCR was performed using iTaq Universal One-Step RT-qPCR Kits (Bio-Rad, cat. # 1725150). RT-qPCR primers were designed using Primer3 (*Koressaar and Remm, 2007*) and are listed in *Supplementary file 13*. The results were normalized as the percentages of 16 rRNA.

## Reversion of transposon-insertion mutants and growth curves

To check the reverting events of transposon insertion mutants, Mut41, Mut_UCI22, Mut_UCI43, and Mut_UCI44 were cultured in replicates in LB medium with or without ertapenem (1 µg/ml) were set up and diluted every day. Each day, an aliquot of culture (10 µl) from each strain/replicate were diluted and plated on LB agar plates to quantify cell numbers. Colony PCR was performed in 24 randomly selected colonies for PCR amplification of the *ompK36* locus, including 500 bp upstream and 100 bp downstream regions. The PCR product was run in agarose gels to assess the size and subsequently Sanger sequenced. One revertant from Mut41 was used for subsequent growth experiment and RT-qPCR to measure the expression of *ompK36*. Growth of UCI38, Mut41, and Mut41_revertant was monitored in a Tecan plate reader in LB medium at 37°C for 8 hr. All experiments were repeated three times.

## Conjugation

Rifampin mutants of UCI38, Mut34, Mut101, Mut186, and Mut195 were raised by plating the exponential growth phase cells on LB agar plates containing 50 µg/ml rifampin. After overnight incubation, rifampin mutants from each strain were selected and subjected to WGS. Mutants that only have mutations in *rpoB* were selected for conjugation. Exponential growth phase cells of rifampin mutants from these five strains were mixed with BIDMC45 cells at 1:1 ratio, then the mixture was spotted on LB agar plates without antibiotics or containing meropenem (0.003 µg/ml) and grown overnight. The second-day morning, cells were transferred to LB liquid medium, serial diluted, and plated on LB agar plates containing meropenem (2 µg/ml) and rifampin (50 µg/ml) for the selection of conjugants. Meanwhile, diluted cells were plated on rifampin (50 µg/ml) plates to quantify cell concentrations. All experiments were repeated three times.

## Acknowledgements

We thank James Gomez for instructions and suggestions of setting up the modified Luria-Delbrück experiment and his further input and comments on this project. We thank Noam Shoresh for the valuable discussions on the data analysis of the modified Luria-Delbrück. We thank Sharon Wong, Anne Clatworthy and Thulasi Warrier for their comments on the manuscript. This publication was supported in part by the National Institute of Allergy and Infectious Diseases of the National Institutes of Health under award 5R01AI117043-05 to DTH and U19AI110818 to the Broad Institute, and by a generous gift from Anita and Josh Bekenstein.

## Additional information

### Funding

| Funder | Grant reference number | Author |
|---|---|---|
| National Institute of Allergy and Infectious Diseases | 5R01AI117043-05 | Deborah Hung |
| National Institute of Allergy and Infectious Diseases | U19AI110818 | Ashlee M Earl |

The funders had no role in study design, data collection and interpretation, or the decision to submit the work for publication.

### Author contributions

Peijun Ma, Conceptualization, Data curation, Software, Formal analysis, Validation, Investigation, Visualization, Methodology, Writing - original draft, Writing - review and editing; Lorrie L He, Hannah H Laibinis, Investigation; Alejandro Pironti, Data curation, Software, Formal analysis, Visualization, Writing - review and editing; Christoph M Ernst, Roby P Bhattacharyya, Resources, Writing - review and editing; Abigail L Manson, Data curation, Software, Formal analysis, Writing - review and editing; Ashlee M Earl, Conceptualization, Resources, Supervision, Writing - review and editing; Jonathan Livny, Formal analysis; Deborah T Hung, Conceptualization, Resources, Supervision, Funding acquisition, Writing - original draft, Writing - review and editing

## Author ORCIDs

Peijun Ma (iD) https://orcid.org/0000-0002-7670-7016
Alejandro Pironti (iD) https://orcid.org/0000-0002-7759-6776
Christoph M Ernst (iD) https://orcid.org/0000-0002-5575-1325
Abigail L Manson (iD) http://orcid.org/0000-0002-3800-0714
Roby P Bhattacharyya (iD) http://orcid.org/0000-0001-6955-5088
Ashlee M Earl (iD) https://orcid.org/0000-0001-7857-9145
Deborah T Hung (iD) https://orcid.org/0000-0003-4262-0673

## Decision letter and Author response

Decision letter https://doi.org/10.7554/eLife.67310.sa1
Author response https://doi.org/10.7554/eLife.67310.sa2

---

# Additional files

## Supplementary files

- Supplementary file 1. Bacterial strains and plasmids used in this study.
- Supplementary file 2. Number of beta-lactamase genes in 267 *K. pneumoniae*.
- Supplementary file 3. CRISRP-Cas systems and R-M systems in 267 *K. pneumoniae*.
- Supplementary file 4. The presence or absence of CRISPR-Cas systems in 2453 *K. pneumoniae* strains.
- Supplementary file 5. Spacer sequences in MGH21.
- Supplementary file 6. Prevalence of spacer-hit genes in plasmid sequences.
- Supplementary file 7. ResFinder results of plasmids containing spacer11-hit gene.
- Supplementary file 8. ResFinder results of plasmids containing spacer24-hit gene.
- Supplementary file 9. Numbers of mutants with mutations (SNP/Short INDEL vs. TN insertion) in *ompK36*.
- Supplementary file 10. ISs involved in inactivating *ompK36* in UCI38.
- Supplementary file 11. MICs and fold changes of MICs measured from 90 mutants.
- Supplementary file 12. Validation of mutations causing decreased susceptibility.
- Supplementary file 13. Primers used in this study.
- Transparent reporting form

## Data availability

All data generated or analyzed during this study are included in this article and in the supplementary files. Sequencing data is deposited to NCBI under the accession number PRJNA670748.

The following dataset was generated:

| Author(s) | Year | Dataset title | Dataset URL | Database and Identifier |
|---|---|---|---|---|
| Ma P, He LL, Pironti A, Laibinis HH, Ernst CM, Manson AL, Bhattacharyya RP, Earl AM, Livny J, Hung DT | 2020 | Genetic determinants facilitating the evolution of carbapenem resistance in Klebsiella pneumoniae | https://www.ncbi.nlm.nih.gov/bioproject/670748 | NCBI BioProject, PRJNA670748 |

The following previously published dataset was used:

| Author(s) | Year | Dataset title | Dataset URL | Database and Identifier |
|---|---|---|---|---|
| Cerqueira GC, Earl AM, Ernst CM, View ORCID Profile, Grad YH, Dekker JP, Feldgarden M, Chapman SB, Reis-Cunha JL, Shea JB, Young S, Zeng Q, Delaney ML, Kim D, Peterson EM, O'Brien TF, Ferraro MJ, Hooper DC, Huang SS, Kirby JE, Onderdonk AB, Birren BW, Hung DT, Cosimi LS, Wortman JS, Murphy CI, Hanage WP | 2017 | Carbapenem Resistance | https://www.ncbi.nlm.nih.gov/bioproject/202876 | NCBI BioProject, PRJNA202876 |

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
