## [Decision Letter]

**Acceptance summary:**

In this work, Ma et al. investigate the breadth of genetic mechanisms for evolution of carbapenem resistance across various lineages of the bacterial pathogen *Klebsiellapneumoniae*. The authors performed systematic and thorough bioinformatic and genetic analyses to identify how transposon activity and CRISPR-Cas systems facilitate the evolution of antibiotic resistance and restriction of horizontally acquired genetic elements, respectively. The study's results emphasize the importance of additional factors, other than MIC values, such as genetic background, plasmid/transposon activity, and drug identity and choice in determining the rate at which resistance can evolve.

**Decision letter after peer review:**

Thank you for submitting your article "Genetic determinants facilitating the evolution of resistance to carbapenem antibiotics" for consideration by *eLife*. Your article has been reviewed by 2 peer reviewers, and the evaluation has been overseen by a Reviewing Editor and George Perry as the Senior Editor. The following individual involved in review of your submission has agreed to reveal their identity: Camilo Barbosa (Reviewer #2).

Essential revisions:

Both reviewers thought the work is important and well performed and have made several straightforward-to-address comments that would improve the manuscript's clarity. The sole important concern raised by both reviewers is that you cast your work and present its findings as if they are clinically actionable, but evidence for this is lacking in your manuscript. Thus, you will either need to connect your work to clinical actions or state this gap as a limitation / future direction of the study. If you take the second option, please revise the Introduction and Discussion sections of your manuscript accordingly.

Reviewer #2 (Recommendations for the authors):

I would consider putting Supplementary figure 1 with the description of the adjusted fluctuation assay method in the main document.

In the same lines, I would consider simplifying figure 1 and more strongly highlight the properties of the 10 selected isolates. As it stands, it is overwhelming to try to identify all of the properties of the strains.

For simplicity I would additionally considering giving the strains a shorter, generic name (Strain A) and have a table with all the relevant information. This could help to read the paper in a simpler way.

---

## [Author Response]

Reviewer #2 (Recommendations for the authors):I would consider putting Supplementary figure 1 with the description of the adjusted fluctuation assay method in the main document.

We have moved this figure to Figure 1B.

In the same lines, I would consider simplifying figure 1 and more strongly highlight the properties of the 10 selected isolates. As it stands, it is overwhelming to try to identify all of the properties of the strains.

Thank you for the suggestion. In the revision, we added a new table (Table 1) to summarize relevant genetic features of these ten isolates selected for this study.

For simplicity I would additionally considering giving the strains a shorter, generic name (Strain A) and have a table with all the relevant information. This could help to read the paper in a simpler way.

Thank you for the suggestion. However, as the names of these isolates have been published previously, we felt it important to use their previously used strain names. (Reference 8 (PMID: 28096418).) The genome information of these isolates can also be found by searching with these names in NCBI isolate browser (https://www.ncbi.nlm.nih.gov/pathogens/isolates/). We hope the addition of Table 1 will help readers navigate through the manuscript a bit more easily.